# A Comprehensive Analysis of the Role of Oxidative Stress in the Pathogenesis and Chemoprevention of Oral Submucous Fibrosis

**DOI:** 10.3390/antiox11050868

**Published:** 2022-04-28

**Authors:** Luciano Saso, Ahmad Reza, Emily Ng, Kimtrang Nguyen, Sheng Lin, Pangzhen Zhang, Paolo Junior Fantozzi, Guliz Armagan, Umberto Romeo, Nicola Cirillo

**Affiliations:** 1Department of Physiology and Pharmacology “Vittorio Erspamer”, Sapienza University of Rome, P. le Aldo Moro 5, 00185 Rome, Italy; luciano.saso@uniroma1.it; 2Melbourne Dental School, The University of Melbourne, 720 Swanston Street, Carlton, Melbourne, VIC 3053, Australia; arrez@student.unimelb.edu.au (A.R.); emilyn1@student.unimelb.edu.au (E.N.); kimtrangn@student.unimelb.edu.au (K.N.); lins4@student.unimelb.edu.au (S.L.); 3School of Agriculture and Food, Faculty of Veterinary and Agricultural Sciences, The University of Melbourne, Parkville, Melbourne, VIC 3052, Australia; pangzhen.zhang@unimelb.edu.au; 4Department of Oral and Maxillofacial Sciences, Sapienza University of Rome, Via Caserta, 6, 00161 Rome, Italy; paolojunior.fantozzi@uniroma1.it (P.J.F.); umberto.romeo@uniroma1.it (U.R.); 5Department of Biochemistry, Faculty of Pharmacy, Ege University, Bornova, Izmir 35100, Turkey; guliz.armagan@ege.edu.tr

**Keywords:** oral submucous fibrosis, oxidative stress, reactive oxygen species, antioxidant supplements

## Abstract

Oral submucous fibrosis (OSMF) is a chronic oral potentially malignant disorder (OPMD). It is described as a scarring disease of the oral mucosa associated with excess oxidants and insufficient antioxidants. While it is becoming increasingly accepted that oxidative stress results in excessive accumulation of collagen and progressive fibrosis of the submucosal tissues, there is limited data regarding the moderation of oxidative stress to initiate or prevent OSMF. To assess the scope for mechanism-based approaches to prevent or reverse OSMF, we systematically evaluated the existing literature and investigated the role of oxidative stress in the pathogenesis and chemoprevention of OSMF. A search for relevant articles on PubMed and Scopus was undertaken using pre-defined inclusion and exclusion criteria. A total of 78 articles were selected in accordance with Preferred Reporting Items for Systematic Reviews and Meta-Analyses extension for Scoping Reviews (PRISMA-ScR) guidelines. The articles eligible for assessment investigated both OSMF and/or oxidative stress biomarkers or specific antioxidants. Both in vitro and human studies consistently demonstrated variations in oxidative stress biomarker levels in OSMF and revealed an increase in oxidative stress, paralleling the development of the disease. Furthermore, the use of antioxidant supplements was overall associated with an improvement in clinical outcomes. Having identified the significance of oxidative stress in OSMF and the therapeutic potential of antioxidant supplements, this scoping review highlights the need for further well-designed studies in the development of mechanism-based interventions for managing OSMF.

## 1. Introduction

Oral submucous fibrosis (OSMF) is a chronic and subtle oral potentially malignant disorder (OPMD) affecting the oral cavity and oropharynx, in mainly the South-Asian populations [1]. Clinically, signs and symptoms of OSMF include but are not limited to, trismus, restricted tongue protrusion, burning mouth, marble-like appearance of the oral mucosa, xerostomia, recurrent ulceration, tongue papillae atrophy, and presence of palpable fibrous bands [2]. These oral complications are fundamentally due to a juxtaepithelial inflammatory reactions leading to fibroelastic changes of the lamina propria, and subsequently to the stiffness of the oral mucosa, which significantly affects patients’ quality of life by reducing their ability to eat and speak. 

Although OSMF is multifactorial in origin, there is overwhelming evidence that long-term areca-nut chewing (alone or in a mixed package known also as betel quid (BQ)), is considered the main etiological factor. Constituents of BQ have in fact been shown to generate substantial amounts of reactive oxygen species (ROS) [3], which may create a biological imbalance between oxidants and antioxidants [4], playing a significant role in OSMF pathogenesis through an excessive accumulation of free radicals and production of lipid peroxides (LPO) [5,6]. BQ chewing is a time-honoured tradition for 10–20% of the world’s population [7]; therefore, attempting to eradicate a habit that has been passed down through countless generations may not be realistic in the short term. Rather than imposing a change in culture and way of life, a novel approach may be to elucidate ways to transpose oxidative stress pathways in favour of antioxidants, to prevent or reverse OSMF. While oxidative stress in OSMF unequivocally results in excessive collagen accumulation and marked fibrosis, a comprehensive review of the role of oxidative stress and antioxidant pathways in initiating and preventing OSMF is yet to be considered. 

The aim of this study was to evaluate the existing literature systematically to assess the role of oxidative stress in OSMF. It was concluded that excess oxidants may drive the pathogenesis, while an excess of antioxidants may play a fundamental role in the chemoprevention of OSMF and the potential reversal of its debilitating effects.

## 2. Materials and Methods

### 2.1. Protocol and Search Strategy

This scoping review was done according to Preferred Reporting Items for Systematic Reviews and Meta-Analyses extension for Scoping Reviews (PRISMA-ScR) guidelines. A search for relevant articles on Pubmed and Scopus published up to June 2021 was completed using the search string: ‘(“oral submucous fibrosis” OR betel OR “piper betel” OR “areca nut” OR gutka OR paan OR “pan masala” OR “slaked lime”) AND (“Reactive Oxygen Species” OR “oxidat*” OR “free radicals” OR “ROS” OR “Superoxide Dismutase” OR “Catalase” OR “Glutathione peroxidase” OR “antiox*” OR “Lipid Peroxides”)’. 

### 2.2. Inclusion and Exclusion Criteria

Inclusion criteria included studies with OSMF human subjects, in vivo or in vitro models of OSMF, as well as studies assessing oxidative stress biomarkers or molecules or using antioxidants in models of fibrosis. Exclusion criteria included non-English language and non-peer-reviewed studies, systematic reviews, meta-analyses, and books/book chapters.

### 2.3. Study Selection and Data Extraction

The selection process involved three stages (Figure 1); the search string was imputed into databases, inclusion and exclusion criteria were applied, and duplicates were manually removed; then, papers obtained from stage 1 were screened based on titles and abstracts according to the inclusion and exclusion criteria under the guidance of the senior author (NC); full texts were analysed according to the inclusion and exclusion criteria. During all the stages, manuscripts that held any uncertainty regarding their relevance to aims of this scoping review were brought to the senior author (NC) and discussed collegially to come to a final decision. 

Data extracted from each article were the name of the first author, publication year, study type (in vitro, in vivo, human), PICO, oxidative stress biomarkers (if present), study design or model, assay methodology, details of treatment/intervention, if any. 

## 3. Results

### 3.1. Overview of the Search Process

Details of the selection process are shown in Figure 1. A total of 549 and 489 (*n* = 1038) articles were retrieved from Scopus and PubMed, respectively. After removing 300 duplicates, the datasets were combined, giving a total of 738 manuscripts. A further 614 papers were excluded due to inconsistent titles and abstracts, leaving a total of 124 articles. After reading the full text, a further 45 manuscripts were excluded. The remaining 78 articles were deemed to be eligible for detailed assessment, of which 52 were laboratory-based studies (Table 1), 27 were clinical studies (Table 2), and 1 was a combination of the above-mentioned ones. There was often an overlap between clinical and laboratory-based studies, and these were included in the category that best represented the study. For example, the assessment of glutathione in a patient’s serum or saliva in the absence of clinical outcome measures was classified as a laboratory-based study.

### 3.2. Laboratory-Based Studies

Of the 51 laboratory-based studies and 1 dual study [8] included in this scoping review, 14 (26.9%) investigated the pathogenesis of OSMF, 13 (25%) demonstrated the significance of oxidative stress in the pathogenesis of OSMF, 7 (13.5%) investigated other biomarkers present in OSMF, 9 (17.3%) investigated antioxidant enzyme status in OSMF patients versus healthy controls, and ultimately 9 (17.3%) examined potential therapeutics for OSMF. 

#### 3.2.1. Pathogenesis of OSMF (Role of Betel Nut/Arecoline, Copper, and Eugenol)

Areca nut extracts (ANE) have been shown to have cytotoxic effects on hamster cheek pouch [3], normal mucosal cells [9,10], and gingival keratinocytes [11,12]. Consumption of BQ has been suggested to be associated with OSMF in oral epithelial cells [13,14], normal buccal mucosa fibroblasts (BMFs) [15], keratinocytes [16], gingival fibroblasts [17], HaCaT and HPL1D epithelial cell lines [18], and human umbilical vein endothelial cells [19]. Furthermore, copper has been shown to enhance the cytotoxicity of arecoline on epithelial cells [20], while eugenol is involved in the pathogenesis of OSMF in oral mucosal fibroblasts [21].

#### 3.2.2. Oxidative Stress Biomarkers in the Pathogenesis of OSMF 

Arecoline induces ROS generation and cell cycle arrest in human keratinocytes [22]. Compared to healthy subjects, OSMF patients had elevated levels of serum malondialdehyde (MDA) [8,23,24,25,26], copper [27,28], LPO, ceruloplasmin [28,29], nitric oxide [30], calcium, magnesium, potassium, iron, conjugated dienes, and hydroxyl radicals [28]; as well as elevated levels of salivary 8-Hydroxy-2′-deoxyguanosine (8-OHdG) [31], 8-isoprostane [32,33], lactoperoxidase, and total protein [34]. Conversely, OSMF patients had relatively decreased levels of hydrogen peroxide (H_2_O_2_) and sodium, compared to healthy controls [28].

On the other hand, there are conflicting results from the nine studies (17.3%) comparing antioxidant enzyme status in OSMF patients versus their healthy counterparts. Of these, three found decreased levels [28,35,36], while six found increased levels [8,23,34,37,38,39] of antioxidant enzymes. Banerjee and colleagues (2020) [40] evaluated mitochondrial antioxidant levels and found higher GSH and peroxiredoxins 3 levels, as well as lower glutaredoxin 2 and catalase levels. Five articles (55.5%) found more significant fluctuations with higher OSMF clinical grading and staging. Specifically, Sadaksharm (2018) [30] found lower levels of superoxide dismutase in OSMF samples, and Gupta and colleagues (2004) [8] found decreased serum β-carotene and vitamin E antioxidant levels. Lee and co-workers (2016) [41] found higher levels of transglutaminase-2 (TGM-2) expression. 

#### 3.2.3. Other Molecular Markers in OSMF

In OSMF serum samples, there were increased platelets, eosinophils, and erythrocyte sedimentation rates, and decreased haemoglobin, iron, ceruloplasmin, copper, and zinc levels, which were well correlated with disease progression [42]. There were also decreased levels of ascorbic acid and increased levels of fatty acids (FAA) [43], serum citrate, oxaloacetate, 8-OHdG [44], and 8-iso-prostaglandin F2 alpha [43]. Other markers in OSMF include tissue inhibitors of metalloproteinases (TIMP-1 and TIMP-2) [45], cyclophilin A [46], and serum uric acid level [47].

#### 3.2.4. Potential Therapeutic Agents for OSMF 

Chang et al. (2012) [48] found that N-Acetyl-L-Cysteine (NAC), apoptosis signal-regulating kinase 1 inhibitor thioredoxin, and c-Jun NH2-terminal kinase inhibitor SP600125 significantly reduced thrombin-induced connective tissue growth factor (CCN2) synthesis in human Bcl2-modified factors, while epigallocatechin-3-gallate (EGCG) completely inhibited thrombin-induced CCN2 synthesis. EGCG was later proposed as a potential therapeutic agent for OSMF [49,50,51,52]. In addition, EGCG, glutathione, and NAC were shown to be effective in inhibiting IL-6-induced epithelial-mesenchymal transition by ANE [53]. Other proposed antioxidants for managing OSMF include lycopene [54], curcumin [55], and angiotensin 1–7 [56].

### 3.3. Clinical Studies

#### Chemopreventive Effects of Nutrient Antioxidants as a Potential OSMF Treatment

A total of 26 clinical studies and one dual study [8] supported the chemoprotective role of antioxidant supplementation. Clinical improvements in at least one OSMF symptom (mouth opening, mucosal burning sensation, tongue protrusion, cheek flexibility, difficulty in swallowing and speech, pain associated with the lesion, oral health-related quality of life) were observed after administration of lycopene [57,58,59,60,61,62,63,64,65], alpha-lipoic acid [66], allicin [67], rebamipide [68], pentoxifylline [69,70], oxitard [59,71,72], aloe vera [71,73,74,75,76,77], curcumin [77,78,79,80,81], and spirulina [75,82] (Table 2).

## 4. Discussion

The aim of this review was to evaluate the existing literature and assess the role of oxidative stress in the pathogenesis and chemoprevention of OSMF. Overall, both laboratory-based and clinical studies consistently demonstrated variations in oxidative stress biomarker levels in OSMF, highlighting their important roles in OSMF pathogenesis and their potential as diagnostic, prognosis, or therapeutic biomarkers. Administration of nutrient antioxidants is a potentially efficacious treatment for OSMF by having chemopreventive effects with clinical improvement. 

### 4.1. Oxidative Stress Biomarkers Is OSMF 

Serum samples from OSMF patients revealed increased MDA [24], ceruloplasmin [29], LPOs [8], nitric oxide [30], and decreased levels of beta-carotene, vitamin E [8], and SOD levels [30]. Human studies largely demonstrated elevated levels of oxidative stress biomarkers serum MDA [8,23,25,26,83], salivary 8-hydroxy-2′-deoxyguanosine [31], salivary 8-isoprostane [33], salivary lactoperoxidase and total salivary protein [34], and serum copper [27,28], calcium, magnesium, potassium, iron, LPOs, conjugated dienes, and hydroxyl radicals [28]. MDA was found by two articles to be more significant with higher clinical OSMF staging and grading [8,83], while 8-OHdG (salivary oxidative stress biomarker) levels were almost double in OSMF patients [31]. Interestingly, studies investigating the level of antioxidant enzymes in OSMF have produced conflicting results. Lower antioxidant enzyme activity in OSMF patients found in some studies [28,35,36] may be due to depletion of antioxidant defence systems occurring as the consequence of overwhelming free radicals by the elevated levels of oxidative stress. Chitra et al. [28] also demonstrated decreased levels of H_2_O_2_ and sodium in OSMF.

Copper (in high levels in AN) initiates fibrinogenesis through the upregulation of lysyl oxidase, thereby inhibiting collagen degradation. High serum copper levels generate high levels of free radicals by metal-catalysed Haber–Weiss reaction, being one of the drivers of carcinogenesis in areca nut users. As a consequence, this compromises blood supply resulting in decreased flow of nutrients and ultimately will impact antioxidant levels. In line with this, Khan et al. (2015) [20] revealed that treating keratinocytes with arecoline and copper resulted in enhanced cytotoxicity, which becomes comparable to IC50 of ANE. 

Glutathione S-transferase (GST), a family of Phase II detoxification enzymes that function to protect cellular macromolecules from attack by reactive electrophiles, are long-time known to protect cells from oxidative stress [84]. However, Bathi, Rao, and Mutalik (2009) [35] and Madhulatha et al. (2018) [36] did not observe strong associations between GST gene polymorphisms (GSTM1 and GSTT1) and OSMF. Compared to healthy controls, OSMF patients had considerably elevated levels of glutathione, ceruloplasmin, and malondialdehyde, but reduced levels of beta-carotene, vitamin E, and glutathione peroxidase (GPx). The three major families of SOD, with the availability of lowering the toxic effects of superoxide (O_2_^−^) and H_2_O_2_ by converting them into water, are copper/zinc, iron/manganese, and nickel type [85], and were found in decreased levels in most studies [24,30,39], but increased in others [28]. Lower antioxidant enzyme activity in OSMF patients may be due to depletion of antioxidant defence systems occurring as a consequence of overwhelming free radicals by the elevated levels of oxidative stress. In particular, increased MDA levels in serum may serve as a valuable surrogate marker in the early diagnosis, treatment, and prognosis of OSMF. Indeed, Gupta et al. (2004) [8] demonstrated that beta-carotene and vitamin E levels in plasma, increased after 6 weeks of their oral administration to OSMF patients, along with decreased MDA levels associated with clinical improvement. 

In summary, there is sound evidence that oxidative stress biomarkers are altered in OSMF tissues as well as in patients’ blood. 

### 4.2. Chemopreventive Effects of Nutrient Antioxidants as an Efficacious Treatment for OSMF 

Multiple compounds with antioxidant and anti-cariogenic properties were investigated to assess their effectiveness in managing or preventing OSMF. The mechanism of action of antioxidants may involve immune system stimulation and breakage of the free radical chain reactions [86].

Six articles investigated the efficacy of aloe vera in managing OSMF with substantial antioxidant vitamins and enzymes, topical aloe vera was effective in improving at least one of the following parameters: mouth burning and opening [71,72,73,74,76,77], tongue protrusion [71,73,76], cheek flexibility [73,76], and speech and swallowing [71]. 

Lycopene is a carotene, carotenoid pigment, and phytochemical with potent antioxidant and anti-carcinogenic properties. Documented as a non-invasive treatment option, it yields significant improvements in OSMF signs and symptoms [54]. Its administration improved mouth opening [57,58,60,61,62,63,64,65], tongue protrusion [58,61], cheek flexibility [61], and burning sensation [58,59,60,61,64,65]. When compared to betamethasone injections, Goel and Ahmed (2015) [57] found that lycopene was more effective in improving mouth opening in subjects with an initial mouth opening distance of less than 19 mm, but less effective with distances between 20–44 mm. 

Curcumin, the main natural polyphenol found in Curcuma species [87], has been shown to target multiple signalling molecules. The current literature suggests that the use of curcumin (either as a topical gel or oral tablets) in OSMF patients improved burning sensation [77,79,80], mouth opening [77,79,80,81], and tongue protrusion [79]. Curcumin was found to be more effective than intralesional steroid injections in improving tongue protrusion and mouth burning [80]. Meanwhile, arecoline can stimulate CCN2, enhancing OSMF’s pro-fibrotic activity. Curcumin can block the arecoline-induced CCN2 expression, thus making it a potentially useful agent in controlling OSMF [16,55]. Furthermore, curcumin influences levels of oxidative stress markers and antioxidants: it decreased MDA and 8-oxo-2′-deoxyguanosine levels [78] while increasing salivary and serum vitamin C/E levels. Vitamin C plays a protective role in carcinogenesis as an antioxidant, reducing vitamin E degradation and enhancing detoxification via cytochrome P450 [88]. 

Epigallocatechin gallate (EGCG) is a plant-based potent antioxidant protecting against cellular damage caused by free radicals [53], with potential uses in OSMF. Hsieh and colleagues (2018) noted that EGCG dose-dependently inhibited arecoline-induced transforming growth factor 1 (TGFb1) activation in BMFs. BMFs exposed to arecoline resulted in the generation of mitochondrial ROS, which activated latent TGFb1, and in turn, stimulated CCN2 and early growth response-1 (Egr-1) synthesis. TGFb1 may play a pivotal role in the pathogenesis of OSMF; thus, EGCG can be a useful agent in the chemoprevention and treatment of OSMF. EGCG blocks TGFb1-induced CCN2 synthesis by suppressing c-JunNH2-terminal kinase (JNK), p38 mitogen-activated protein kinase, and activin receptor-like kinase 5 (ALK5) [49]. Hsieh and co-workers (2017) concluded that ALK5, Smad3, extracellular signal-regulated kinase, and JNK are involved in the TGFb1-induced Egr-1 protein production in BMFs. Egr-1 mediates COL1A1 and COL1A2 mRNA expression and acid-soluble collagen production in BMFs. EGCG can block TGFb1-induced collagen production by attenuating Egr-1 expression, which is a key mediator in the TGFb1-induced pathogenesis of OSMF. Hsieh et al., 2015, noted that arecoline induces an overexpression of Egr-1, which enhances the profibrotic activity seen in OSMF. EGCG was shown to completely block arecoline-induced Egr-1 expression in human buccal fibroblasts. 

Lastly, other compounds such as alpha-lipoic acid (ALA) are worth noting due to their effects in alleviating burning sensation and improving mouth opening. Along with conventional intralesional steroid injections, ALA was indeed able to reverse higher clinical stages [66]. Jiang et al. [67] demonstrated that when compared to triamcinolone acetonide, allicin, a defence molecule derived from garlic exhibited greater and more stable augmentation in mouth opening, alleviation of mucosal burning sensation, and improvement in ‘Oral Health Related Quality of Life’ (OHRQoL) score. Rebamipide, an amino acid derivative of 2-quinolinone used for gastrointestinal mucosal protection was found effective in managing burning sensation [68]. Oral pentoxifylline, as an adjunct to surgical reconstruction, improves mouth opening, burning sensation, and relapse [70]. Oxitard capsules resulted in significant clinical improvements in mouth-opening, tongue protrusion, swallowing, speech pain associated with the lesion, and burning sensation when compared with placebo [72]. Turmeric/black pepper and Nigella sativa improved mouth opening, burning sensation, and SOD levels [89]. Lastly, spirulina, which has multiple antioxidant effects, was shown to improve mouth opening and burning sensation [82]. 

Eight papers compared the chemopreventive effects of nutrient antioxidants against each other. When compared to standard antioxidant capsules, aloe vera produced greater improvements in burning sensation and mouth opening [74], while pentoxifylline showed further reduction in dysphagia and burning sensation [69]. Aloe vera was less effective in improving mouth opening when compared to spirulina [75], and also less effective in improving swallowing, speech, and pain associated with the lesion when compared with oxitard capsules [71]. However, when compared to curcumin, aloe vera brought larger improvements in burning sensation, but the two were equally effective in increasing mouth opening [77]. Three studies compared lycopene to other antioxidants, yielding conflicting results. Saran et al. [58] found lycopene to be more effective than curcumin in improving mouth opening and burning sensation, while Piyush et al. [61] found no differences between the two. Patil et al. [59] found that lycopene was less effective than oxitard in improving mouth opening and tongue protrusion. Compared to Nigella sativa, turmeric/black pepper showed greater improvements in mouth opening, burning sensation, and SOD levels [89]).

In summary, results of clinical studies show that there is clear scope for investigating the preventive and therapeutic effects of antioxidants in betel nut chewers and patients with OSMF through well-designed, large, controlled studies.

### 4.3. Strengths and Limitations

By providing a broad overview of relevant studies, this scoping review has established the current understanding of the topic of interest while identifying gaps in the literature. However, we must acknowledge the possibility that we may have missed some relevant studies including non-English articles and articles with no full text. The balance of breadth and depth of the data reported in this article was also challenging due to the volume of articles identified and time constraints. 

This present review was not aimed at providing a comprehensive pathogenic model of OSMF. While oxidative stress is indeed associated with OSMF, tissue inflammation is crucial for the induction of tissue fibrosis. The involvement of prostanoids and other inflammatory mediators [90,91] as well as immune cells (recently reviewed in [92]) is central to OSMF pathogenesis but was not in the scope of our review. 

### 4.4. Future Directions

The current literature is in favour of antioxidants being used for OSMF prevention and management because it is relatively non-toxic and can be easily supplemented in the diet. Our findings have important public health implications for the management and reversal of OSMF’s debilitating effects. Additionally, our results can be used as a helpful precursor for future systematic reviews to deepen knowledge of pathogenesis and chemoprevention of OSMF. This review may also serve as a framework for future studies to help inform the development of mechanism-based interventions and a clear guidelines for managing patients with OSMF using antioxidants.

## 5. Conclusions

Understanding the molecular profile of distinct BQ components and how these mediate the pathogenesis of OSMF is a key challenge if we are to develop mechanism-based preventive or therapeutic strategies for this potentially malignant disease [93,94]. This scoping review highlighted the role of oxidative stress in OSMF’s pathogenesis, as shown by altered levels of various oxidative stress biomarkers, including MDA and 8-OHdG. As such, there is the potential for these oxidative stress biomarkers to be used in OSMF diagnosis, prognosis, and potential therapeutic targets. Similarly, antioxidant enzymes, (e.g., serum SOD and ceruloplasmin) may also be used in OSMF diagnosis and prognosis as their levels were also changed in OSMF patients. Furthermore, this scoping review identified various nutrient antioxidants, (e.g., aloe vera, lycopene, curcumin, and EGCG) effective in improving the signs and symptoms of OSMF such as mouth opening, burning sensation, tongue protrusion, and cheek flexibility.

## Figures and Tables

**Figure 1 antioxidants-11-00868-f001:**
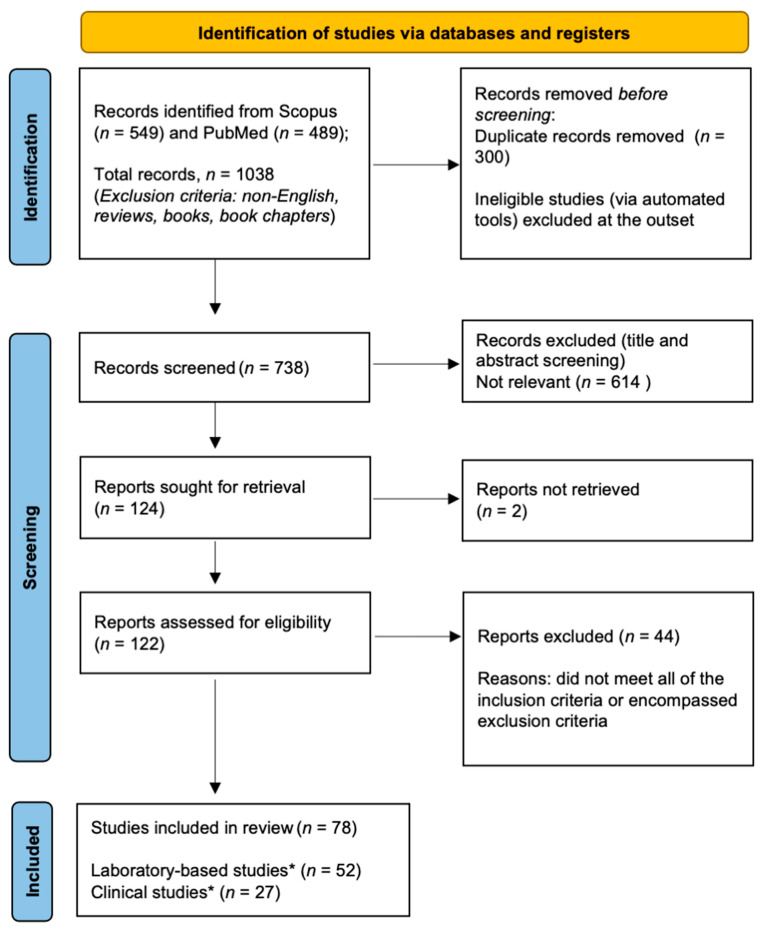
Flow chart of the selection process according to PRISMA-ScR guidelines. Asterisk (*) denotes the inclusion of a dual (laboratory-based and clinical) study.

**Table 1 antioxidants-11-00868-t001:** Overview of laboratory-based studies that were eligible for analysis.

Author/s, Year	Relevant Biomarker (s)	OSMF Samples or Model	Intervention	Control/Comparison	Study Type
Aggarwal et al., 2011	beta-carotene	blood samples from OSMF patients	—	age- and sex-matched controls	biochemical
Anuradha and Devi, 1995	haemoglobin; ceruloplasmin; iron; copper; zinc	blood samples from OSMF patients; staged	—	age- and sex-matched healthy controls	biochemical
Avinash et al., 2014	MDA; SOD	blood samples from OSMF patients	—	healthy subjects	biochemical
Bathi et al., 2009	GSH; ceruloplasmin; MDA; GSTM1; GSTT1	blood samples from OSMF patients	—	age-, sex-, and SES-matched controls	biochemical
Bale et al., 2017	MDA; SOD	blood samples from OSMF patients; staged	—	non-symptomatic OSMF	biochemical
Banerjee et al., 2020	SOD2, Catalase, GLRX2, GSH, GPx, TXN2	mitochondria purified samples taken from OSMF patients	—	mitochondrial antioxidants in healthy controls and OPMD	biochemical
Chang et al., 2001	GSH; H_2_O_2_; mitochondrial membrane potential	oral OSF fibroblasts and oral KB epithelial cells treated with AN extracts and arecoline	—	baseline data (untreated control)	cell culture
Chang et al., 2002	unscheduled DNA synthesis	gingival keratinocytes treated with AN extracts	Vit C, GSH, NAC, Deferoxamine	untreated control, other treatments	cell culture
Chang et al., 2013	TGFβ1-induced CCN2 synthesis	buccal mucosal fibroblasts	EGCG; JNK, p38 MAPK, and ALK5 inhibitors	untreated control	cell culture
Chang et al., 2014	PGE2, COX-2, CYP1A1, HO-1	gingival keratinocytes exposed to AN extracts	piper betle leaf (PBL) extract, hydroxychavicol, dicoumarol, curcumin	untreated control, other treatments	cell culture
Chang et al., 2016	8-isoprostane, IL-1α; ADAM 17; PGE2; COX2	primary human gingival keratinocytes treated with AN extracts and arecoline	α-naphthoflavone, aspirin, catalase; MEK, JAK, and Src inhibitors	untreated control, other treatments	cell culture
Chitra et al., 2012	LPO; conjugated dienes; HO; SOD; H_2_O_2_; copper; calcium; magnesium; potassium; iron	saliva samples from OSMF patients	—	age- and sex-matched healthy controls	biochemical
Deng et al., 2009	CCN2	OSMF tissue samples; normal buccal mucosal fibroblasts treated with arecoline	NAC, curcumin	normal oral mucosa; untreated control	IHC, cell culture
Divyambika et al., 2018	LPO; GSH; SOD; GPx; Vit A, C and E	saliva samples from OSMF patients	—	age- and sex-matched healthy controls	biochemical
Francis et al., 2019	—	OSMF cell lines	lycopene; quercetin	—	cell culture
Gupta et al., 2004 ^	MDA; ROS	blood samples from OSMF patients; graded	—	healthy controls	biochemical
Gurudath et al., 2012	SOD, GPx	blood samples from OSMF patients	—	age- and sex-matched healthy subjects	biochemical
Guruprasad et al., 2014	Vitamin C; iron	blood samples from OSMF patients	—	healthy patients	biochemical
Hou et al., 2017	Cyclophilin A (CYPA) via 2D gel electrophoresis/mass spectrometry	tissue biopsy from OSMF patients	—	normal mucosal tissue	biochemical
Hsieh et al., 2015	Egr1	OSMF tissues; buccal mucosal fibroblasts treated with arecoline	NAC; EGCG; JNK, ERK inhibitors	untreated control	IHC, cell culture
Hsieh et al., 2017	Egr1, COL1A1, COL1A2	buccal mucosa fibroblast cultures stimulated with TGF-β	EGCG; ERK, JNK, p38 MAPK, ALK5, inhibitors	untreated control	cell culture
Hsieh et al., 2018	TGFβ; ROS; CCN2, Egr-1	human buccal mucosal fibroblasts treated with arecoline	EGCG; TGFβ inhibitor; antioxidant	untreated control	cell culture
Illeperuma et al., 2015	ROS; GRO-α, IL6, IL8; DNA double strand breaks, 8-oxoG	OSMF tissues; immortalised human normal oral keratinocytes and AN-exposed fibroblasts	antioxidants; NOX1 and 4 silencing	normal oral mucosa; untreated controls	IHC, cell culture
Jeng et al., 2004	mitochondrial membrane potential depolarization; GSH; ROS	oral KB epithelial cells treated with hydroxychavicol	NAC, SOD, catalase	untreated and DMSO-treated cells	cell culture
Jeng et al., 1994a	GSH, ATP, xanthine oxidase	normal oral mucosal fibroblasts treated with eugenol	—	untreated control	cell culture
Jeng et al., 1994b	DNA strand break	oral mucosal fibroblasts incubated with different BQ constituents	GSH, cysteine, mannitol, catalase, SOD	untreated controls	cell culture
Kapgate et al., 2020	MDA	blood samples from OSMF patients	turmeric	healthy subjects	biochemical
Khan et al., 2015	ROS, catalase activity	human keratinocytes and gingival fibroblasts treated with arecoline and AN extracts	Cu; GSH, SOD, NAC	untreated controls	cell culture
Khanna et al., 2013	copper; zinc; selenium and molybdenum	blood samples from OSMF patients	—	healthy subjects and OSCC patients	biochemical
Kim et al., 2020	Gro-α, IL-6, IL-8; EMT	HPV16 E6/E7-transfected immortalised human oral keratinocytes (IHOK)	EGCG; GSH; NAC	—	cell culture
Kulasekaran et al., 2020	8-OHdG	OSMF tissues (very early, early, moderately advanced, and advanced)	—	normal buccal mucosa	IHC
Lee et al., 2016	ROS	OSMF buccal mucosa biopsy sample; normal oral fibroblasts	GSH; NAC; EGCG	normal buccal mucosa	IHC; cell culture
Li et al., 2019	ROS; PERK; collagen;	OSMF tissues; HHUVECs treated with arecoline; OSMF mouse model	verteporfin	normal buccal mucosa; untreated controls	IHC, cell culture, mouse model
Madhulatha et al., 2018	glutathione GSTM1, GSTT1	blood samples from OSMF patients	—	healthy subjects	biochemical
Meera et al., 2020	8-isoprostane	blood and saliva samples from OSMF patients	—	OSCC and control patients	biochemical
Nair et al., 1992	ROS, DNA damage	Syrian golden hamsters exposed to various BQ components; OSMF patients	—	Untreated controls (no atropine); healthy subjects	Animal model, ICC
Nandakumaar et al., 2020	8-OHdG	saliva samples from OSMF patients	—	age- and sex-matched healthy controls, OSCC patients	biochemical
Pant et al., 2016	TGF-β signaling (ATF2; pJNK); ROS	HaCaT and HPL1D epithelial cell lines exposed to AN extracts; OSMF tissues	—	normal buccal mucosa; untreated controls	cell culture, IHC
Paulose et al., 2016	MDA	blood samples from OSMF patients	—	age- and gender-matched healthy individuals	biochemical
Pitiyage et al., 2012	TIMP-1; TIMP-2	early and advanced OSMF tissues; OSMF fibroblasts	—	age-matched healthy controls and paan users	cell culture, IHC
Rai et al., 2010	MDA, 8-OHdG	blood and saliva samples from OSMF patients	curcumin (1 g)	healthy patients	comparative, biochemical
Rai et al., 2019	8-OHdG; 8-epi-PGF2α; Protein carbonyl	blood samples from OSMF patients	—	serum sample of healthy patients	biochemical
Rathod et al., 2018	beta-carotene	blood samples from OSMF patients	—	age- and gender-matched control	biochemical
Sadaksharam, 2018	NO; SOD	OSMF and OSCC patients	—	healthy controls	biochemical
Senghore et al., 2018	8-OHdG; 8-isoprostane	blood samples from OPMD male patients	—	plasma 8-OHdG and 8-isoprostane levels in OL	biochemical
Shah et al., 2017	ceruloplasmin	blood samples from OSMF patients	—	blood samples from healthy controls	biochemical
Shakunthala et al., 2015	MDA; antioxidant activity	OSMF patients	—	age-matched controls	biochemical
Singh et al., 2015	COX-2	OSMF tissues, fibroblasts from OSMF, and normal oral fibroblasts treated with arecoline	—	healthy subjects; untreated controls	cell culture, IHC
Thangjam and Kondaiah, 2009	heme oxygenase-1; ferritin light chain; G6PDH; GCLC; GSH; IL-1a; p38 MAPK	human keratinocyte cells (HaCaT cell line) treated with arecoline	—	untreated controls	cell culture
Tsai et al., 2009	Heme Oxygenase-1	OSMF tissues; fibroblasts from OSMF and normal oral fibroblasts treated with arecoline	—	normal oral tissues; untreated controls	cell culture
Yadav et al., 2020	uric acid	blood samples from OSMF patients	—	healthy controls, leukoplakia, and OSCC	biochemical
You et al., 2019	AT1R; Mas1; NOX4; IL-Iβ; α-SMA; collagen type 1; CCN2; NLRP3; AT1R; ACE; ACE2; H_2_O_2_	OSMF tissues; animal model of OSMF in ALL Sprague-Dawley rats; normal oral fibroblasts treated with arecoline	VE0991	normal oral tissues; positive and negative controls	cell culture

—, not applicable; α-SMA, Smooth Muscle alpha-Actin; 8-epi-PGF2α, 8-epi-Prostaglandin F2alpha; 8-OHdG, 8-Hydroxydeoxyguanosine; ACE, Angiotensin-Converting Enzyme; ADAMs, A Disintegrin And Metalloproteinase; ALK, Activin Receptor-Like Kinase; AN, Areca Nut, AOA, Aminooxyacetic Acid; AT1R, Angiotensin II Receptor Type 1; ATF2, Activating Transcription Factor 2; CCN2, Connective Tissue Growth Factor; COX-2, Cyclooxygenase-2; CYP1A1, cytochrome P450 1A1; EGCG, Epigallocatechin-3 Gallate; Egr1, Early Growth Response 1; ERK, Extracellular Signal-Regulated Kinase; G6PDH, Glucose 6 Phosphate Dehydrogenase; GCLC, Glutamate-Cysteine Ligase Catalytic subunit; GLRX2, Glutaredoxin 2; GSH, Glutathione; Glutathione Peroxidase, GPx; HO-1, hemeoxygenase-1; H_2_O_2_, Hydrogen Peroxide; HHUVECs, Human Umbilical Vein Endothelial Cells; IHOK, Immortalized Human Oral Keratinocytes; ICC, immunocitochemistry; IL, interleukin; JNK, c-Jun NH2-terminal Kinase; KGM-SFM, Keratinocyte Growth Medium; LPO, Lipid Peroxide; MAPK, Mitogen-Activated Protein Kinase; MDA, Malondialdehyde; NAC, N-Acetyl-L-Cysteine; NO, Nitric Oxide; NOX; NADPH Oxidase; OL, Oral Leukoplakia; OLP, Oral Lichen Planus; OPMD, Oral Potentially Malignant Disorders oculopharyngeal muscular dystrophy; OSCC, Oral Squamous Cell Carcinoma; OSMF, Oral Submucous Fibrosis; ROS-PERK, Reactive Oxygen Species—Protein Kinase RNA-like Endoplasmic Reticulum Kinase; SOD—Superoxide Dismutase; TGFβ, Transforming growth factor beta; TIMP, Tissue Inhibitor Matrix Metalloproteinases; TXN2, Thioredoxin 2. ^ Gupta et al. (2004) is a dual laboratory-based and clinical study.

**Table 2 antioxidants-11-00868-t002:** Overview of clinical intervention studies that were eligible for analysis.

Author/s, Year	Antioxidant (s)	Clinical Parameters	Control/Comparison	Study Type
Anuradha et al., 2017	systemic (juice) and topical (gel) aloe vera	BS; CF; MO; TP	hydrocortisone; hyaluronidase; antioxidant supplements	RCT
Arakeri et al., 2020	lycopene (4 mg/day for 3 months)	BS; MO	placebo capsule	RCT
Baptist et al., 2016	rebamipide (100 mg t.i.d. for 21 days)	BS	betamethasone (4 mg/mL biweekly for 4 weeks)	RCT
Goel and Ahmed., 2015	lycopene capsules (2 mg, b.i.d. for 6 months)	MO	no treatment; betamethasone (4 mg/mL) diluted in 1 mL of 2% xylocaine (biweekly for 6 months)	RCT
Gowda et al., 2011	lycopene capsules with zinc, selenium, and phytonutrients (2000 μg, b.i.d. for 3–6 months)	BS; MO; healing of ulcers; mucosal color/texture	baseline (before treatment); non placebo-controlled	RCT
Gupta et al., 2004 ^	beta-carotene (50 mg); Vit A palmitate (2500 IU); Vit E acetate (10 IU); Vit C; zinc manganese; copper	MO; TP	age- and sex-matched healthy controls; baseline (before treatment)	RCT
Jiang et al., 2015	allicin (1 mg TCM-046, 99% HPLC intralesional injection)	MO, BS	triamcinolone acetonide (intralesional injection)	RCT
Johny et al., 2019	lycopene (8 mg b.i.d for 3 months); lycopene (8 mg b.i.d for 3 months)/hyaluronidase (1500 IU twice/week for 3 months)	MO	placebo capsules	RCT
Kalkur et al., 2014	Pentoxifylline (400 mg/day)	BS, speech	standard antioxidants	RCT
Kapoor et al., 2019	curcumin (400 mg/day for 3 months)	pain, MO	—	RCT
Karemore et al., 2012	lycopene (4 mg b.i.d for 3 months)	MO	placebo capsule	RCT
Kholakiya et al., 2020	pentoxifylline (400 mg t.i.d for 3 months)	MO, BS, malignant transformation, relapse	—	retrospective
Kumar et al., 2007	lycopene (8 mg b.i.d for 6 months); curcumin (300 mg b.i.d for 6 months)	MO, BS	placebo capsule	RCT
Patil et al., 2014	oxitard (2 capsules/day); topical aloe vera (5 mg t.i.d for 3 months)	MO, TP, swallowing, speech, pain	—	RCT
Patil et al., 2015a	xitard (2 capsules b.i.d for 3 months)	MO, TP, BS, pain, swallowing, speech	placebo capsules	RCT
Patil et al., 2015b	spirulina (500 mg/day for 3 months); topical aloe vera (5 mg t.i.d for 3 months)	MO, BS, pain, ulcers/erosions/vesicles	—	RCT
Patil et al., 2018	oxitard (2 capsules t.i.d for 3 months); lycopene (8 mg/day for 3 months)	MO, TP, swallowing, speech, pain, BS	—	RCT
Pipalia et al., 2016	turmeric (400 mg)/black pepper (100 mg) (2 capsules t.i.d for 3 months); nigella sativa (2 × 500 mg capsules t.i.d for 3 months)	MO, BS, CF, TP	—	RCT
Piyush et al., 2019	curcumin (300 mg b.i.d for 6 months); lycopene (8 mg b.i.d for 6 months)	MO, BS, TP, CF	placebo capsules	RCT
Rai et al., 2019	curcumin (300 mg t.i.d for 12 weeks)	MO, BS, TP, adverse reactions	standard antioxidants	RCT
Rajbhoj et al., 202	aloe vera (~5 mg/day); curcumin gel (~5 mg/day)	MO, BS	—	RCT
Rao PK. 2010	alpha lipoic acid (once/day for 3 months) and betamethasone (1 mL) and hyaluronidase (1500 IU); once/week for 12 weeks	MO, BS	betamethasone (1 mL) and hyaluronidase (1500 IU); once/week for 12 weeks	RCT
Saran et al., 2018	lycopene (4 mg/day for 3 months); curcumin (300 mg t.i.d for 3 months)	MO, BS	—	RCT
Shetty et al., 2013	Spirulina (500 mg b.i.d)	MO, BS	placebo capsules	RCT
Singh et al., 2016	Aloe vera	BS, MO, CF, TP	standard antioxidant capsule	RCT
Sudarshan, 2012	topical aloe vera (~5 mg t.i.d for 3 months)	MO, CF, TP	standard antioxidant capsule	RCT
Yadav et al., 2014	curcumin (2 × 300 mg/day for 3 months)	BS, MO, TP	dexamethasone (4 mg) and hyaluronidase (1500 I.U)	RCT

—, not applicable; 8-OHdG, 8-Hydroxydeoxyguanosine; μg, microgram; b.i.d., bis in die; BS, Burning Sensation; CF, Cheek Flexibility; g, grams; GPx, Glutathione Peroxidase; GSH, reduced Glutathione; GSTM1, Glutathione S-transferase Mu 1; GSTT1, Glutathione S-transferaseTheta 1, H_2_O_2_, HPLC, High-Performance Liquid Chromatography; Hydrogen Peroxide; HO, Hydroxyl radicals; IU, international units; LPO, Lipid Peroxides; MDA, Malondialdehyde; mg, milligrams; ml, millilitre; MO, Mouth Opening; RCT, Randomised Controlled Trial; ROS, Reactive Oxygen Species; SES, Socioeconomic Status; SOD, Superoxide Dismutase; t.i.d., ter in die; TP, Tongue Protrusion; Vit, Vitamin. ^ Gupta et al. (2004) is a dual laboratory-based and clinical study.

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
