# Peer review of "A Comprehensive Analysis of the Role of Oxidative Stress in the Pathogenesis and Chemoprevention of Oral Submucous Fibrosis"

_antioxidants, 2022, doi:10.3390/antiox11050868_

Round 1

Reviewer 1 Report

Dear authors,

We have read your manuscript investigating the role of oxidative stress. The work is very well documented and exhaustive about the topic. 

As you stated in your manuscript, you provide a broad overview of relevant studies, thus establishing the current understanding of the topic of interest, even though they might be some gaps in the litterature.

Laboratory-based and clinical studies demonstrated variations in oxidative stress biomarker levels in OSMF, highlighting their roles as biomarkers. And administration of nutrient antioxidants is a potentially efficacious treatment with chemopreventive effects and clinical improvement. 

The overall work is a bit heavy to read, as there are so many reagents investigated...

Remark: "A search for relevant articles on Pubmed and Scopus was done from April to July 2021". Instead of indicating the period when you did your investigations, please indicated the considered period for articles publication.

And, please, check your research string (as I find 309 articles on PubMed, when I copy-Paste it, instead of the 489 that you claim).

Therefore, we recommend to accept your manuscript for publication, after these slight modifications.

Author Response

Please see document attached

Reviewer 2 Report

This is a potential interesting manuscript dealing the role of oxidative stress on the pathogenesis of OSMF and raised some possible future therapeutic strategies. Some revisions are suggested.

  1. While some papers are not included for this systemic review for oxidative stress in oral submucous fibrosis, they can be included for discussion for the manuscript (e.g., Int J Mol Sci. 2020 Oct 30;21(21):8104 and perhaps some papers from the study groups of Prof. Bartsch & Nair, Prof. Warnakulasuriya, or Professor PA Reichart etc.).
  2. While oxidative stress is associated with OSMF, tissue inflammation (page 2, 1st paragraph) is especially crucial for the induction of tissue fibrosis. The involvement of prostanoids (PGE2, etc.) and other inflammatory mediators such as IL-6, IL-8 or others by betel quid components should be discussed and referenced.
  3. Page 2, 2nd paragraph: “……..marked fibrosis, a comprehensive review of the role of oxidative stress in initiating and preventing OSMF is yet to be considered “. Why oxidative stress, but not antioxidants are able to prevent OSMF?
  4. Page 4, last paragraph (section 3.2.2), some controversial results. The authors may give some possible explanation for this point.
  5. Small conclusion at the end of paragraph can be given for some long sections.
  6. Abbreviations should be spelled out when they first appear in the text.
  7. Some typewriting errors should be checked and corrected through the text.
  8. For study on curcumin, the authors may also check the studies of Prof. Shun-Fa Yang.
  9. References 9 and 21: 1994a, 1994b. The authors may delete a and b.

Round 2

Reviewer 2 Report

improved much. accept.